# Effects of LED Light Quality on the Growth of Pepper (*Capsicum* spp.) Seedlings and the Development after Transplanting

Xiaojuan Liu, Rui Shi, Meifang Gao , Rui He, Yamin Li  and Houcheng Liu *

College of Horticulture, South China Agricultural University, Guangzhou 510642, China
* Correspondence: liuhch@scau.edu.cn; Tel.: +86-20-8528-0464

**Abstract:** In this study, the effects of different light conditions on the growth of pepper seedlings and the development of these pepper seedlings after transplanting were explored. Based on the control light, pepper seedlings were grown under radiation with different proportions of added blue, UV-A, and far-red light for 18 days. Compared with the control, supplementation with UV-A and far-red light increased the seedling height whereas blue light decreased. Blue and UV-A treatments increased seedling compactness and the seedling index while far-red light treatments have obvious inhibitory effects. The chlorophyll content of the UV-A treated seedlings was significantly increased, but far-red light reduced the carotenoid content. Far-red light increased the activities of SOD and CAT and decreased the MDA content of seedlings. After transplanting, there was no obvious difference in the flowering time of all treated pepper plants. An increase in pepper yield was discovered when pepper seedlings were supplemented with blue and UV-A light. Overall, our results demonstrated that proper supplementation of UV-A and blue light at the seedling stage positively produces strong and healthy pepper seedlings, and could increase the yield of pepper plants after transplanting.

**Keywords:** growth; light spectra; pepper seedlings; the seedling index



## 1. Introduction

As one of the important environmental factors, light plays essential roles in all physiological processes of plant growth and development [1,2]. Through the aspects of light intensity, light photoperiods, and light quality, the morphological and physiological traits of plants are affected. The solar light spectrum, spanning from UV to far-red, can be perceived by plants and has been widely investigated for its role in the growth and development of plants. For example, far-red light influences physiological processes such as seed germination, hypocotyl growth, induction of flowering, etc. [3]. It has been demonstrated that red light is beneficial to increasing the leaf area and promoting the accumulation of photosynthetic products [4]. Blue light is mainly involved in the regulation of stomatal movement and photosynthetic performance [5,6]. Green light inhibited stem growth and chloroplast gene expression of plants, which were opposite to blue or red light [7]. In addition, UV-A has a positive role in accumulating healthy compounds in different species [8,9].

Pepper (*Capsicum* spp.) is one of the worldwide grown vegetables whose growth and development are largely influenced by different light wavelengths. The anthocyanin content and the fruit ripening time of purple pepper were influenced by the increase of blue light proportion when white light was the basal light [10]. Under exposure to UV-B light, the vegetative growth of pepper plants was inhibited but the flavonoid accumulation in the leaves increased [11]. Taller pepper plants with greater dry weight and thicker stems were induced by the higher green light proportion, which might be due to improved photosynthetic efficiency and increased photosynthesis [12]. Moreover, pepper plants radiated under distinct light spectra showed different vulnerabilities to water deficits [13].

Hence, altering the light environment according to the response of peppers to different light qualities could be used as an approach to cultivating peppers with desired traits.

Seedling growth is a critical developmental stage of vegetable growth and is the basis for high-quality vegetable production [14]. Plant factories with artificial light (PFALs) are indoor farms that are important vegetable seedling and production systems [15]. It has been reported that cultivating vegetable seedlings in PFALs has some advantages, such as a short cultivation time and high energy efficiency [16,17]. For pepper seedling growth, researchers have revealed the influence of blue and far-red light on seedlings' development [18,19], but it remains to be further explored which light quality or which ratio of light quality is the most beneficial for the growth of healthy and strong pepper seedlings. Moreover, few studies have described the flowering and fruiting traits of pepper seedlings treated with different light qualities after transplanting to plastic greenhouses or outdoors. And it was found that few articles have reported the impacts of UV-A on the performance or characteristics of pepper seedlings until now.

To compare the roles of specific light conditions on the growth and development of pepper plants, in this study, we studied the impacts of a different portion of blue, UV-A, and far-red light spectrums on the growth and development of pepper seedlings under a similar PPDF of 200 $\mu molm^{-2}s^{-1}$ and the flowering and fruiting characteristics of these pepper seedlings treated with different conditions after transplanting. Meanwhile, the Pearson analysis and principal component analysis were utilized to comprehensively select the favorable light quality for healthy and strong pepper seedling production, which provided a reference for the cultivation of strong pepper seedlings in plant factories by altering the light environments.

## 2. Materials and Methods

### 2.1. Plant Material and Growth Conditions

Seedlings were cultivated in an artificial plant factory of South China Agriculture University (Guangzhou, China) with a constant temperature (24 ± 2 °C) and (humidity of 75 ± 5% as well as air carbon dioxide concentration of approximately 565 ± 10 ppm. The pepper seeds (cv Xiangyan No. 55, Hunan Xiangyan Seed Industry Co., Ltd.) were sown in sponge cubes (2 cm × 2 cm × 2 cm) and were planted in the hydroponic after germination. After seed germination, the white light with a photosynthetic photon flux density (PPFD) of 200 $\mu molm^{-2}s^{-1}$ was used for pretreatment for 2 days. White: red = 3:2 was proved to be more conducive to pepper breeding than single white light in our previous study. Based on this, the seedlings were divided into 7 groups to grow under seven different light treatments CK:3W2R (W:R = 3:2); B30:3W2R + 30 $\mu molm^{-2}s^{-1}$B; B60:3W2R + 60 $\mu molm^{-2}s^{-1}$B; UVA2:3W2R + 2 $\mu molm^{-2}s^{-1}$UV-A; UVA6:3W2R + 6$\mu molm^{-2}s^{-1}$UV-A; FR20:3W2R + 20 $\mu molm^{-2}s^{-1}$FR; FR30: 3W2R + 30 $\mu molm^{-2}s^{-1}$FR. The photosynthetic photon flux density (PPFD) of all treatments was set as 200 $\mu molm^{-2}s^{-1}$ (Figure 1, Table S3), and the photoperiod was 12/12 h (day/night). The adjustable LED panels (Chenghui Equipment Co., Ltd., Guangzhou, China; 150 cm × 30 cm; white light: 400–700 nm; red light: 660 ± 10 nm; UV-A light: 385 ± 10 nm; and FR light: 735 ± 10 nm) were used as light sources. The PPFD and light spectra were measured by a spectroradiometer (ALP-01, Asensetek, Taiwan). The nutrient solution used during the experiment was a half-strength Hoagland nutrient solution (pH 5.5) with an electrical conductivity (EC) of 1.3 mScm$^{-2}$. After 18 days of light treatment, all the pepper seedlings grew to more than six true leaves and were transplanted into a plastic greenhouse (South China Agriculture University, Guangzhou, China).

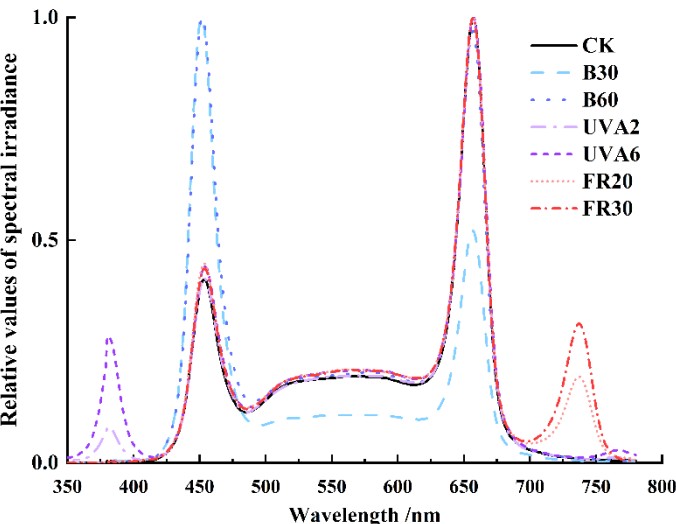

**Figure 1.** The spectral profile of CK, B30, B60, UVA2, UVA6, FR20, and FR30 treatments are delivered by different combinations of LEDs.

*2.2. Determination of Morphological Traits and Fresh and Dry Weight*

Fifteen uniform seedlings were selected to determine their growth indexes. For daily growth rate determination, measurements were taken 3 times every 6 days after the first day of light treatment. The seedling height and hypocotyl length were measured by using a measuring ruler while the stem diameter was detected by using a vernier caliper, respectively. The growth rate of seedling height was obtained by the following formula:

Growth rate of seedling height = (The seedling height at 18 DAT − the seedling height before treatment)/(Treatment days).

Hypocotyl and stem diameter growth rates were obtained in the same way as above. After 18 days of treatment, the true leaf area was measured by a leaf area meter (LI-3000A). The leaf amounts of seedlings were counted. All the samples were dried at 105 °C for 30 min and then dried at 75 °C to a constant weight. The fresh weight and the dry weight were weighed with a thousandth electronic balance (BCE224-1CCN, Sartorius, Beijing, China).

*2.3. Calculation of Seedling Comprehensive Index*

The comprehensive seedling indexes were calculated according to the following equations:

specific leaf weight = leaf dry weight/leaf area,

the plant compactness = shoot dry weight/plant height,

root to shoot ratio = root dry weight/shoot dry weight,

shoot dry matter content = shoot dry weight/shoot fresh weight,

seedling index = (stem diameter/seedling height + root dry weight/shoot dry weight) · whole plant dry weight [20].

*2.4. Measurement of Pigment Contents and Chlorophyll Fluorescence Parameters*

The fresh leaves of pepper seedlings (0.5 g) were taken for pigment content determination, and the procedures and the pigment content calculation were as described by He et al. [8]. Briefly, samples were immersed in 8 mL of acetone alcohol mixture (acetone: alcohol = 1:1, *v/v*) in the dark until all samples turned white. The absorbance at 663 nm, 645 nm, and 440 nm was determined by a UV-spectrophotometer (Shimadzu UV-16A, Shimadzu Corporation, Kyoto, Japan). For the examination of chlorophyll fluorescence

parameters, a fluorometer (MINI-PAM-II, Germany) was used. The parameters Fv/Fm, Y(II), and ETR in the third true leaf were measured.

### 2.5. Measurement of Phytochemical Substance and Enzyme Activities of Seedlings

Using the Coomassie Brilliant Blue method, the procedures used to determine the total soluble protein content in the experiment were suggested by He et al. [8]. For enzyme activity determination, a fresh leaf of 0.5 g was ground in 10 mL phosphate buffer (50 mM, pH 7.8), centrifuged at $10,000 \times g$ rpm for 20 min, and then the supernatant was the enzyme extract taken for the determination of SOD, CAT, and POD activities. The methods were suggested by Li et al. [21]. A 50 μL enzyme extract was added to the reaction mixture (contained 1.5 mL phosphate buffer, 0.3 mL methionine (130 mM/L), 0.3 mL NBT (750 μM/L), 0.3 mL riboflavin (20 μM/L), 0.3 mL EDTA-Na$_2$ (100 μM/L), and 0.25 mL distilled water) to determine the SOD activity. The mixed reaction solution was terminated after 20 min of 4000 lux illuminations, then the absorbance of the reaction solution was determined at 560 nm using a spectrophotometer. The reaction mixture of POD contained 0.8 mL enzyme extraction, 1.45 mL phosphate buffer, 0.5 mL guaiacol (50 mM/L), 0.5 mL H$_2$O$_2$ (2%), and was measured at 470 nm. The absorbance of the reaction mixture containing 0.2 mL enzyme extraction, 1.5 mL phosphate buffer, and 200 mM H$_2$O$_2$ was monitored at 240 nm.

The MDA content was measured following the methods reported in Zhang et al. [22]. 0.5 g fresh sample was homogenized in a 10 mL 10% trichloroacetic acid (TCA) solution, centrifuged at $4000 \times g$ rpm for 10 min, and the supernatant after centrifugation was taken as the sample extract. After centrifugation, 2 mL supernatant was mixed with 2 mL of 0.6% TBA solution, then the mixture was reacted in boiling water for 15 min. After rapid cooling and centrifugation, the supernatant was measured at 532 nm, 600 nm, and 450 nm using a UV-spectrophotometer.

### 2.6. Data of Flowering and Fruit Set after Transplanting

After 18 days of light treatment, 24 uniform pepper seedlings of each treatment were transplanted into a plastic greenhouse. The substrate was coconut bran. Water and fertilizer were supplied by a drip irrigation system, and the solution was 1/2 Yamazaki pepper nutrient solution. The first flowering time, the node of the first flower, fruits per plant at 15 and 30 days after transplanting (DAT), number of harvested fruits, and total fruit weight were recorded. The number of fruits per plant was calculated by fruit length over 3 cm at the 54 DAT.

### 2.7. Statistical Analysis

All the experimental data were processed and analyzed by the SPSS 26.0 (IBM, Armonk, NY, USA) and Origin 2021 software (Northampton, MA, USA). The Duncan's multiple range test at $p < 0.05$ was used to determine significant treatment differences for all investigated traits. All the pictures were obtained by the Origin 2021 software.

## 3. Results

### 3.1. Impacts of Diverse LED Light Conditions on the Morphology and Growth of Pepper Seedlings

The morphological indexes of the pepper seedlings were significantly influenced after 18 days of treatment. Seedling height in the FR30, FR20, UVA2, and UVA6 treatments were significantly 86%, 53.6%, 10.6%, and 8.6% higher than in the control, respectively, whereas B30 and B60 treated seedlings were 5.1% and 17.5% shorter than the control seedlings (Figure 2A,D). The hypocotyl length of seedlings treated with FR30, UVA2, UVA6, and FR20 significantly increased compared to those of B30, B60, and control, among which FR30 treated seedlings increased by 30.3% (Figure 2B). Compared with the control, FR20 treatment significantly increased the stem diameter by 5.5%, while B60 treatment significantly decreased the stem diameter by 7.7% (Figure 2C). The amounts of true leaves

under each treatment increased significantly compared to the control, with the increase of UVA2 and UVA6 treatments reached 14.7% (Figure 2E).

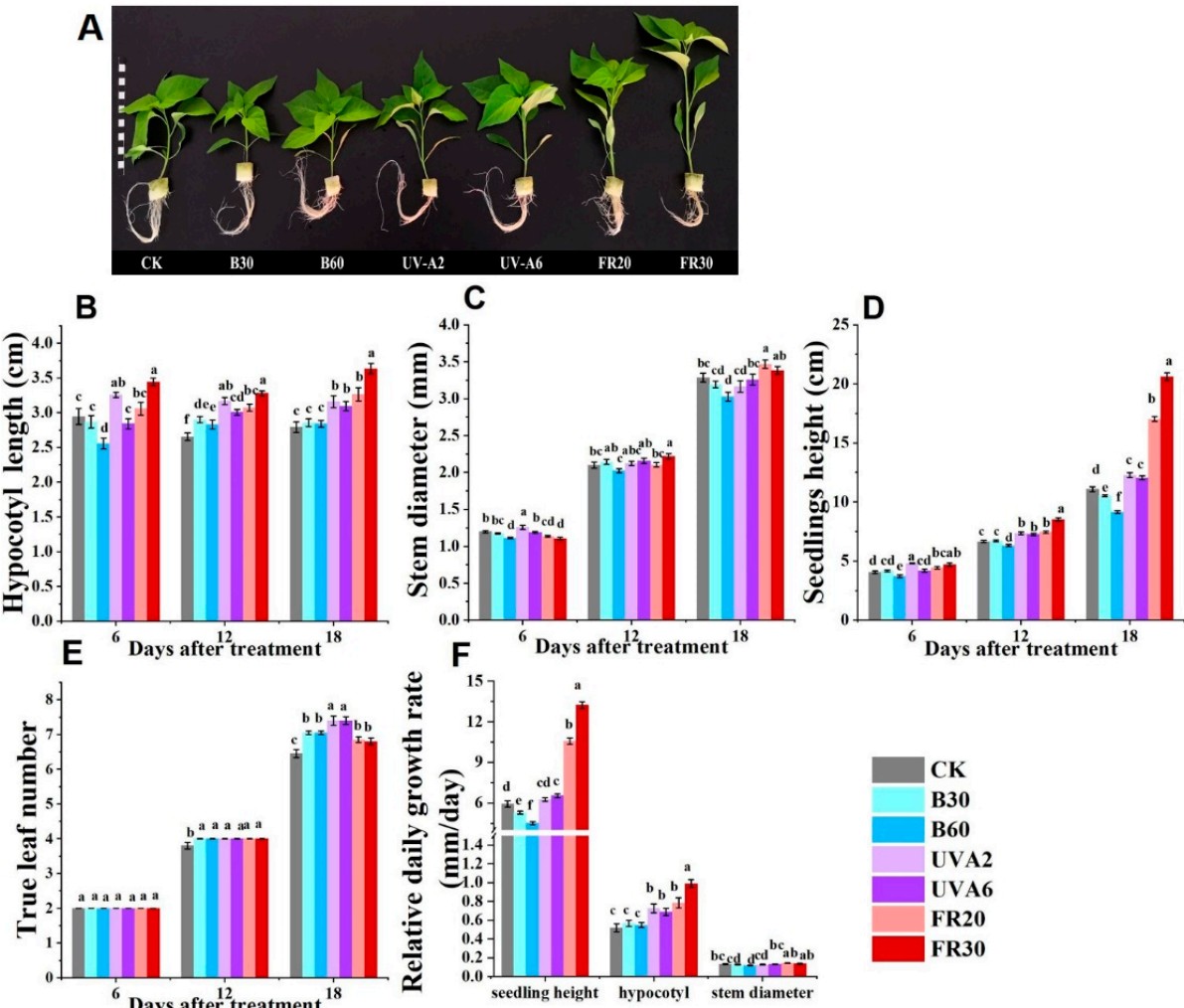

**Figure 2.** Effects of different light conditions on the morphology and growth of pepper seedlings. (**A**) The typical morphology of pepper seedlings radiated under different light conditions for 18 days. Each small square on the left of the picture represents 1 cm × 1 cm. (**B**) Hypocotyl length, (**C**) Stem diameter, (**D**) Seedling height, and (**E**) True leaf number of pepper seedlings treated with distinct light environments. (**F**) The relative daily growth rate of pepper seedlings exposed to various light conditions. Different letters indicate significant differences by Duncan's test at *p* < 0.05.

The relative daily growth rate of pepper seedlings under various light conditions was examined. The seedling height growth rate of seedlings in UVA6, FR20, and FR30 was higher, while those of B30 and B60 were lower than the control, respectively (Figure 2F). The UVA2, UVA6, FR20, and FR30 treatments induced the hypocotyl growth rate 39.6%, 32.4%, 50.8%, and 90.4% higher than the control, respectively (Figure 2F).

There were significant differences in the biomass of pepper seedlings under different light quality treatments. The seedlings grown under UVA2, UVA6, FR20, and FR30 treatments displayed elevated shoot fresh weight, with an increase of 12.8%, 16.1%, 19.1%, and 26.7%, respectively, compared with the control (Table 1). Both the plant fresh weight and the shoot dry weight of seedlings treated with UVA6, FR20, and FR30 significantly increased (Table 1). The FR20 and FR30 treatments led to a significant reduction in root dry weight compared with the control (Table 1). The plant dry weight of UVA6 and FR30 radiated seedlings was significantly 7.6% and 6.5% higher than the control (Table 1). The percentage of dry matter in shoots under B60 and UVA6 treatments was significantly higher

than those of the control by 6.0% and 5.8%, while those of the FR treatment showed no comparable difference (Table 1).

**Table 1.** Effects of different light quality treatments on biomass of pepper seedlings.

| Treatment | Fresh Weight (g/per Plant) | | | Dry Weight (g/per Plant) | | | Percentage of Shoot Dry Matter (%) |
|---|---|---|---|---|---|---|---|
| | Shoot | Root | Whole | Shoot | Root | Whole | |
| CK | 4.62 ± 0.18 c | 2.52 ± 0.12 ab | 7.13 ± 0.21 cd | 0.31 ± 0.01 cd | 0.45 ± 0.01 a | 0.75 ± 0.02 bc | 6.53 ± 0.13 bc |
| B30 | 4.49 ± 0.11 c | 2.28 ± 0.07 b | 6.77 ± 0.17 d | 0.29 ± 0.01 d | 0.44 ± 0.01 ab | 0.73 ± 0.01 c | 6.67 ± 0.05 ab |
| B60 | 4.34 ± 0.12 c | 2.27 ± 0.05 b | 6.61 ± 0.16 d | 0.29 ± 0.01 d | 0.43 ± 0.01 abc | 0.72 ± 0.01 c | 6.89 ± 0.09 a |
| UVA2 | 5.21 ± 0.21 b | 2.43 ± 0.08 ab | 7.64 ± 0.27 bc | 0.34 ± 0.017 bc | 0.44 ± 0.01 ab | 0.78 ± 0.02 ab | 6.58 ± 0.10 bc |
| UVA6 | 5.37 ± 0.19 ab | 2.46 ± 0.07 ab | 7.82 ± 0.25 ab | 0.36 ± 0.01 ab | 0.45 ± 0.01 a | 0.81 ± 0.02 a | 6.88 ± 0.06 a |
| FR20 | 5.50 ± 0.16 ab | 2.42 ± 0.06 ab | 7.92 ± 0.19 ab | 0.35 ± 0.01 ab | 0.43 ± 0.01 bc | 0.78 ± 0.01 ab | 6.35 ± 0.07 c |
| FR30 | 5.86 ± 0.21 a | 2.56 ± 0.11 a | 8.42 ± 0.28 a | 0.38 ± 0.02 a | 0.42 ± 0.01 c | 0.80 ± 0.02 a | 6.43 ± 0.07 bc |

Data represent mean ± SE ($n$ = 20). Different letters indicate significant differences between treatments at the $p < 0.05$ using the Duncan's test.

In summary, the more the proportions of blue light (B30 and B60 treatments), the greater the inhibitory effect on the stem and hypocotyl elongation of pepper seedlings. Both UVA2 and UVA6 treatments increased the biomass, the percentage of shoot dry matter, and true leaf amount. Both FR20 and FR30 treatments were beneficial for increasing hypocotyl growth, stem diameter, and biomass of seedlings.

### 3.2. Influences of Diverse LED Light Conditions on the Comprehensive Growth Indexes of Pepper Seedlings

Since distinct light conditions affected the morphology and biomass of pepper seedlings, we studied the effects of different light environments on comprehensive growth indexes of pepper seedlings. The plant compactness increased in the B60 treatment but was significantly reduced in FR-treated seedlings (Table 2). Seedlings growing under FR20 and FR30 displayed a significantly decreased root-shoot ratio compared with the control (Table 2). The B60 treated seedlings showed obviously higher specific leaf weight, while seedlings of B30, FR20, and FR30 had apparently lower specific leaf weight in comparison with the control (Table 2). Among all the pepper seedlings, only the FR20 and FR30 treated seedlings showed a significantly lower seedling index than those of the control (Table 2).

**Table 2.** The comprehensive index of pepper seedlings of all treatments.

| Treatment | Compactness | Root-Shoot Ratio | Specific Leaf Weight | The Seedling Index |
|---|---|---|---|---|
| CK | 0.03 ± 0.00 b | 0.27 ± 0.02 a | 1.53 ± 0.03 b | 0.210 ± 0.014 b |
| B30 | 0.03 ± 0.00 b | 0.30 ± 0.01 a | 1.33 ± 0.02 c | 0.242 ± 0.009 a |
| B60 | 0.03 ± 0.00 a | 0.32 ± 0.03 a | 1.65 ± 0.02 a | 0.248 ± 0.007 a |
| UVA2 | 0.03 ± 0.00 b | 0.28 ± 0.02 a | 1.48 ± 0.03 b | 0.236 ± 0.010 a |
| UVA6 | 0.03 ± 0.00 ab | 0.27 ± 0.01 a | 1.50 ± 0.03 b | 0.246 ± 0.006 a |
| FR20 | 0.02 ± 0.00 c | 0.21 ± 0.01 b | 1.22 ± 0.02 d | 0.182 ± 0.008 c |
| FR30 | 0.02 ± 0.00 c | 0.19 ± 0.02 b | 1.18 ± 0.02 d | 0.153 ± 0.007 d |

Different letters indicate significant differences between treatments at the $p < 0.05$ using the Duncan's test.

Therefore, an increasing proportion of blue light (B30 and B60 treatments) was conducive to cultivating healthy and strong pepper seedlings with compact phenotypes. The pepper seedlings radiated by UVA2 and UVA6 showed compact seedlings, increased root-shoot ratio, and enhanced the seedling index. Supplementary FR (FR20 and FR30) treatments induced slender pepper seedlings, implying that FR light was negative for cultivating stronger pepper seedlings.

### 3.3. Effects of Different LED Light on Photosynthetic Pigment Content and Chlorophyll Fluorescence Parameters

Wondering whether the photosynthetic traits of pepper seedlings were influenced, the contents of photosynthetic pigment, including chlorophyll and carotenoids, in pepper seedlings were examined. When exposed to UVA2 and UVA6 treatments, the contents of

chlorophyll a were significantly higher than those in control. However, only the seedlings in the UVA6 treatment showed significantly higher total chlorophyll content than those in the control (Table 3). The carotenoid content of FR20 treated seedlings was significantly lower than that of the control, while there was no significant difference between the other treatments and control (Table 3).

**Table 3.** Photosynthetic characteristics and pigment contents of pepper seedlings under different treatments.

| Treatment | Photosynthetic Pigment Content (mg/g FW) | | | | Fv/Fm | Y(II) | ETR |
|---|---|---|---|---|---|---|---|
| | Chl a | Chl b | Total chl | Carotenoids | | | |
| CK | 1.14 ± 0.04 cd | 2.52 ± 0.12 ab | 1.51 ± 0.05 bc | 0.23 ± 0.00 ab | 0.75 ± 0.00 a | 0.59 ± 0.01 ab | 8.89 ± 0.26 ab |
| B30 | 1.20 ± 0.01 bc | 2.28 ± 0.07 b | 1.59 ± 0.01 ab | 0.23 ± 0.00 ab | 0.75 ± 0.00 a | 0.59 ± 0.01 ab | 8.86 ± 0.19 ab |
| B60 | 1.13 ± 0.02 cd | 2.27 ± 0.05 b | 1.50 ± 0.02 bc | 0.22 ± 0.00 ab | 0.76 ± 0.00 a | 0.60 ± 0.02 a | 9.13 ± 0.58 a |
| UVA2 | 1.31 ± 0.05 a | 2.43 ± 0.08 ab | 1.61 ± 0.09 ab | 0.24 ± 0.01 a | 0.75 ± 0.00 a | 0.59 ± 0.01 ab | 8.83 ± 0.19 ab |
| UVA6 | 1.23 ± 0.02 a | 2.46 ± 0.07 ab | 1.65 ± 0.026 a | 0.23 ± 0.00 ab | 0.75 ± 0.00 a | 0.60 ± 0.01 a | 9.05 ± 0.32 a |
| FR20 | 1.06 ± 0.04 d | 2.42 ± 0.06 ab | 1.42 ± 0.05 c | 0.21 ± 0.01 c | 0.75 ± 0.00 a | 0.57 ± 0.01 b | 8.52 ± 0.29 b |
| FR30 | 1.12 ± 0.02 cd | 2.56 ± 0.11 a | 1.51 ± 0.03 bc | 0.22 ± 0.00 bc | 0.76 ± 0.00 a | 0.58 ± 0.01 ab | 8.81 ± 0.27 ab |

Different letters indicate significant differences between treatments at the $p < 0.05$ using the Duncan's test.

Chlorophyll fluorescence parameters, including Fv/Fm, Y(II), as well as ETR, showed no comparable difference among all the treatments in pepper seedlings (Table 3). However, both the Y(II) and ETR of B60, UV-A6 pepper seedlings were slightly higher than those of FR20 pepper seedlings (Table 3). This means that compared to far-red light, blue and UV-A light were better at improving the actual photosynthetic efficiency and electron transport rate of PSII in pepper seedlings.

### 3.4. The Influence of Different LED Light Environment on Physiological Characteristics and Antioxidant Enzyme Activities of Pepper Seedlings

The contents of soluble protein and MDA, as well as the activities of antioxidant enzymes (SOD, POD, CAT), were detected in pepper seedlings. There was no significant difference in soluble protein content among all seedlings (Figure 3A). The SOD activity of UVA2 and UVA6 pepper seedlings was significantly lower than that of the control, decreased by 23.9% and 23.3%, respectively (Figure 3B). No significant difference in POD activities among all the treated seedlings was observed (Figure 3C). The CAT activities increased in the seedlings of FR treatments compared to the control, meanwhile, the MDA content in seedlings of FR20 and FR30 significantly decreased (Figure 3D,E).

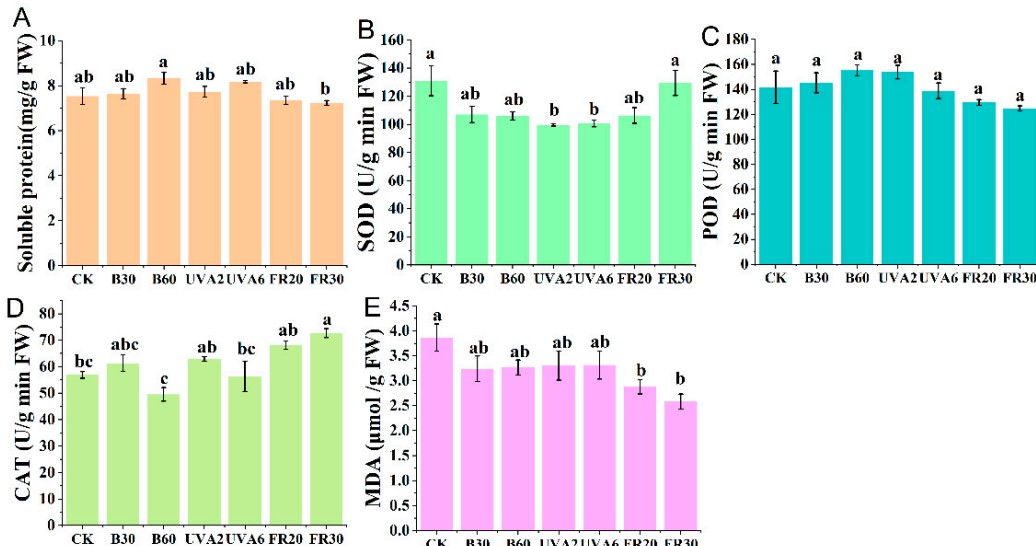

**Figure 3.** The soluble protein content and antioxidant enzyme activities in pepper seedlings of different treatments in this study. (**A**) Soluble protein content, (**B**) SOD, (**C**) POD, (**D**) CAT, and (**E**) the MDA content. Different letters indicate significant differences by Duncan's test at $p < 0.05$.

### 3.5. The Effect of Different LED Light on Flowering and Fruit Setting of Different Seedlings after Transplanting

The growth status of the seedling stage often affects the flowering and fruiting of plants after transplanting [19]. In this study, the first flowering occurred within 14–16 DAT in all treatments (Table 4). However, the first flowering nodes of the FR20 seedlings were significantly higher than those of the control (Table 4). Although no significant difference in the fruit amount among the pepper plants was observed from the 15 to 30 DAT, the fruit amount was the highest in the UVA2 treated pepper plants whereas the lowest in the FR30 treated pepper plants (Table 4). At the 53 DAT, the fruit numbers per plant of B60 radiated pepper plants were significantly 22.5% higher than that of the control, but there was no significant difference among other pepper plants (Table 4). In addition, the significant increase in the average fruit weight showed that UVA6 treated pepper plants showed promoted fruit growth (Table 4). Overall, in the early stage of pepper seedlings, adding a certain proportion of blue light or a certain intensity of UV-A light had positive effects on the development of pepper fruits after transplantation.

**Table 4.** The flowering and fruiting characteristics of all treated pepper seedlings after transplanting.

| Treatment | Time of First Flower (days) | The Node of First Flower | Number of Fruits on the 15th Day | Number Of Fruits At The 30th Day | Average Number of Fruits per Plant | Average Weight of Fruits per Plant (g) |
|---|---|---|---|---|---|---|
| CK | 15.3 ± 0.3 a | 8.1 ± 0.1 bc | 0.2 ± 0.1 ab | 3.1 ± 0.5 ab | 5.0 ± 0.3 b | 13.8 ± 0.6 bc |
| B30 | 14.7 ± 0.3 a | 7.8 ± 0.1 c | 0.2 ± 0.1 ab | 2.5 ± 0.4 ab | 5.2 ± 0.3 b | 16.5 ± 1.9 ab |
| B60 | 14.8 ± 0.3 a | 8.0 ± 0.2 bc | 0.2 ± 0.1 ab | 3.2 ± 0.5 ab | 6.1 ± 0.2 a | 14.6 ± 0.5 abc |
| UVA2 | 14.8 ± 0.3 a | 7.9 ± 0.1 c | 0.3 ± 0.1 a | 3.8 ± 0.4 a | 5.4 ± 0.3 ab | 15.0 ± 0.7 abc |
| UVA6 | 14.9 ± 0.3 a | 8.1 ± 0.1 bc | 0.2 ± 0.1 ab | 2.7 ± 0.3 ab | 5.6 ± 0.3 ab | 17.3 ± 1.2 a |
| FR20 | 15.3 ± 0.3 a | 8.6 ± 0.1 a | 0.2 ± 0.1 ab | 2.5 ± 0.4 ab | 5.3 ± 0.3 ab | 11.7 ± 1.0 c |
| FR30 | 15.5 ± 0.3 a | 8.4 ± 0.2 ab | 0.0 ± 0.0 b | 2.1 ± 0.4 b | 5.1 ± 0.3 b | 11.9 ± 1.2 c |

Different letters indicate significant differences between treatments at the $p < 0.05$ using the Duncan test.

### 3.6. Correlation Analysis and Comprehensive Evaluation

To analyze the correlation between the examined indexes, and to make a comprehensive evaluation of pepper seedlings grown under different light conditions, the Pearson correlation and principal component analysis (PCA) were performed. The Pearson correlation analysis was used to analyze the correlation between each growth and physiological index of pepper seedlings. There was a highly positive correlation between the seedling index and specific leaf weight, root-shoot ratio, plant compactness, chlorophyll a content, chlorophyll b content, total chlorophyll content, the carotenoid content, and POD activity. The seedling index was negatively correlated with shoot dry weight, whole dry weight, shoot fresh weight, whole fresh weight, stem diameter, plant height, flowering time, SOD activity, and CAT activity (Figure 4A). In addition, no correlation was found between the seedling index and the other examined indicators (Figure 4A).

Principal component analysis (PCA) was used to determine the correlation between each index and treatment, with PC1 explained 24.20% and PC2 displayed 18,71% of the total variance (Figure 4B). Positive correlations existed between the seedling index and specific leaf weight, root-shoot ratio, soluble protein content, carotenoid content, chlorophyll a content, percentage of shoot dry matter content, POD activity, and compactness. While it was negatively related to shoot dry weight, whole plant dry weight, shoot dry weight, whole plant fresh weight, stem diameter, plant height, flowering time, SOD activity, and CAT activity (Figure 4B). According to the results, the physiological growth characteristics of pepper seedlings were different under distinct light quality treatments. The seedling growth performance of B30, B60, UVA2 and UVA6 treatments was small, while that of FR20 and FR30 treatments was significantly different from other treatments (Figure 4B).

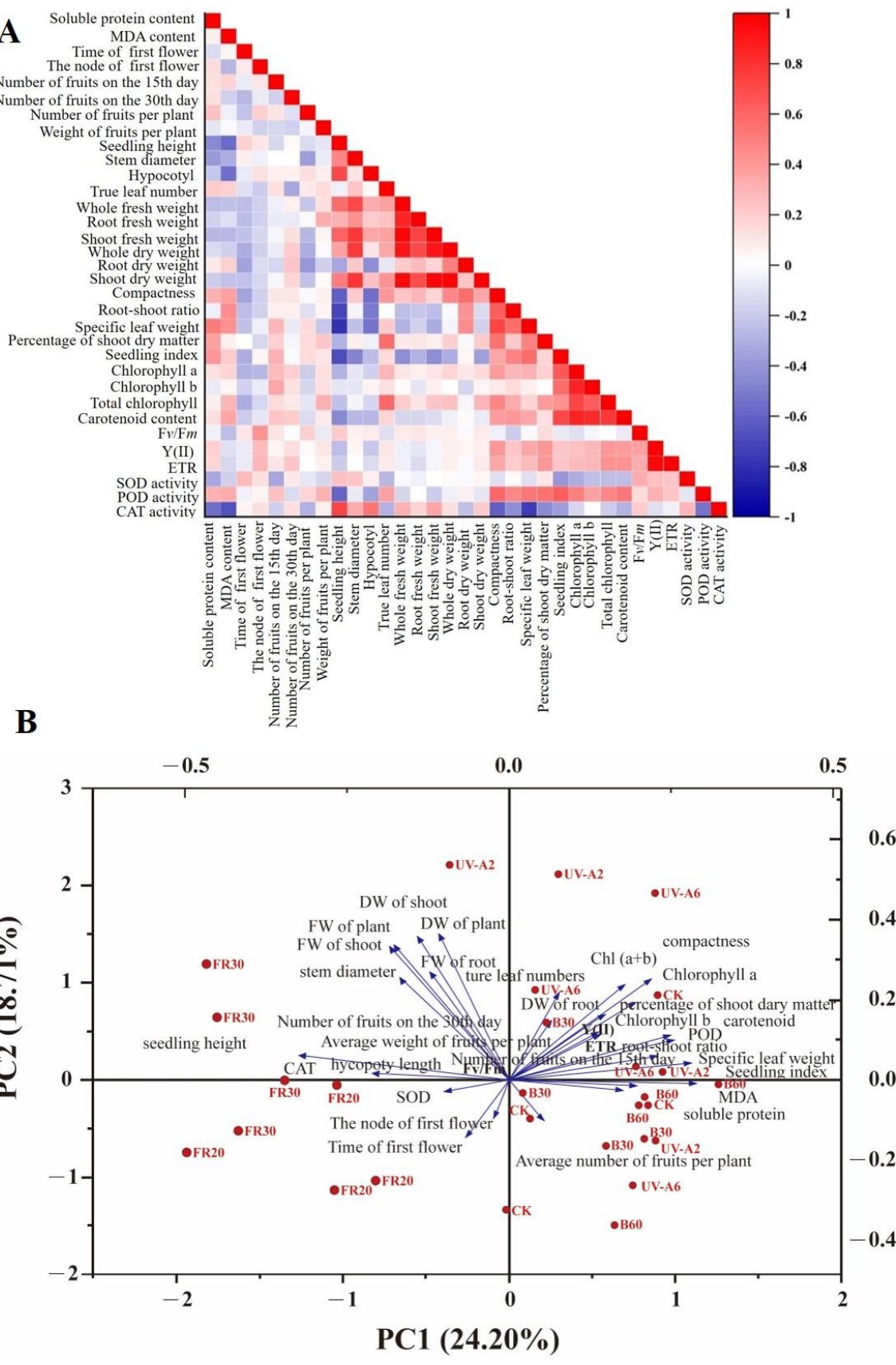

**Figure 4.** Correlation analysis of the examined indexes of pepper seedlings. (**A**) The Pearson correlation of each growth and physiological index of pepper seedlings. (**B**) Principal component analysis of each growth index and treatment of seedlings.

Following the methods of Huang et al. [23] and Sabzalian et al. [24], OriginPro 2021 software was used to simplify these evaluated indicators, then we constructed a comprehensive score model for evaluating the growth and development of pepper seedlings with different light qualities as Y = 25.43% × Y1 + 21.88% × Y2 (The meanings of Y1 and Y2 were listed in the Supplementary Tables S1 and S2). Using this model, the comprehensive score value of the growth and quality index of pepper seedlings was calculated. The higher the comprehensive score value, the better the seedling performance under the treatment. Following the results, the optimal light quality treatment for the growth and quality of pepper seedlings was UVA2 > UVA6 > B60 > B30 > CK > FR30 > FR20.

## 4. Discussion

### 4.1. Impacts of Light Quality on the Growth Traits

When subjected to different light qualities in a PFAL, pepper seedlings showed distinct morphology and growth traits. The seedling height and hypocotyl length of pepper seedlings were promoted by increases in UV-A (2 and 6 $\mu molm^{-2}s^{-1}$) and far-red (20 and 30 $\mu molm^{-2}s^{-1}$) light proportions, whereas inhibited in the supplemental blue light treatments (30 and 60 $\mu molm^{-2}\,s^{-1}$) (Figure 1). Hwang et al. [19] reported that properly supplemental far-red light increased the seedling height of tomato and pepper seedlings, but the seedling height of these seedlings was lower than those of the control as the far-red light exceeded the appropriate ratio. However, the seedling height of cucumber and watermelon seedlings in the added far-red light treatments increased [19]. These indicated that the effects of far-red light on plant height depended on plant species and light intensity. Kang et al. found that the plant height of tomato seedlings grown under appropriate UV-A radiation increased [25], which is similar to our results. Some studies have discovered that blue light inhibited seedling height, which was mainly due to plant cryptochrome activating the expression of genes related to growth inhibition and affecting hormone synthesis [26–28].

The seedling index could fully reflect the growth quality of pepper seedlings. Seedlings with higher compactness are short and denser, often have a higher seedling index, and are more suitable for cultivating high-quality vegetables [29]. Similarly, our PCA results showed that the seedling index was positively correlated with compactness, and negatively correlated with the seedling height (Figure 4). In this study, the plant compactness of B60 and UVA6 seedlings increased, but those of the FR20 and FR30 seedlings decreased (Table 2). Besides, the seedling index of B30, B60, UVA2, and UVA6 treated seedlings was higher than that of the control, whereas those of FR20 and FR30 seedlings were significantly lower (Table 2). These results indicated that different light qualities might influence seedling quality by affecting plant architecture, in addition, supplementary with a specific dose of UV-A and blue light was useful for cultivating healthy and strong pepper seedlings.

### 4.2. Effects of Light Quality on Photosynthetic Pigment Content and Photosynthetic Characteristics

Carotenoids and chlorophyll are mainly photosynthetic pigments, chlorophyll is mainly responsible for the absorption, transmission, and transformation of light energy, while carotenoids function in the capture of light energy and light damage defense [30,31]. Our quantitative analysis displayed that the seedlings of FR treatment displayed a reduction in carotenoid content, while B and UV-A treated seedlings showed no comparable difference with those of the control in carotenoid content (Table 3). In terms of chlorophyll content, our results discovered that the total chlorophyll content showed an increase in seedlings of B30, UVA2 and UVA6 treatments, while those in FR20 seedlings decreased slightly (Table 3). Consistently, the results from a previous study showed that the total chlorophyll content of pepper seedlings in the supplementary blue light (L1.5) treatment, in which the R:B ratio is similar to the R:B ratio in B30 in this study, increased than those of the L3.5 treatment whose R:B ratio is similar with the R:B ratio in this control treatment [32]. UV-A radiation also increased the chlorophyll content of radish plants, which may be the

result of seedlings promoting light capture by increasing chlorophyll content to reduce oxidative stress damage caused by UV-A [33]. The slight reduction of photosynthetic pigment content in seedlings caused by adding FR treatment might be because that FR induces plant elongation and larger leaf area, resulting in decreased pigment content per unit area of the leaves [34].

Studies have proved that leaf pigments are critical for seedling photosynthesis and are key parameters for seedling quality [35,36]. In our study, the PCA results showed that the seedling index was positively correlated with chlorophyll content and carotenoid content. In accordance with this, UV-A treatment significantly increased photosynthetic pigment content in pepper seedlings, which might contribute to enhancing peppers' photosynthesis and seedling quality (Figure 4). The increase of photosynthetic pigment content in pepper seedlings under blue light treatments was less than those under UV-A treatments, which was consistent with the conclusion calculated by the formula that the promoting effects of blue light treatments on seedling quality were less than those of UV-A treatments (Table 3).

Representing the utilization of light energy during leaf photosynthesis, the chlorophyll fluorescence of plants is influenced by light quality [37,38]. In this study, although the photosynthetic pigment content of seedlings treated with different light qualities was altered, we found that the chlorophyll fluorescence parameters, including Fv/Fm, Y(II), and ETR, exhibited no significant difference among all seedlings (Table 3). As light quality regulates photosynthetic characteristics not only by influencing chlorophyll content but also by affecting leaf microstructure [39,40]. The difference in leaf microstructure might account for why there was no significant difference in detected chlorophyll fluorescence parameters of pepper seedlings under different treatments, which needs further exploration.

### 4.3. Effects of Light Quality on Physiological Characteristics of Pepper Seedlings and Transplant Development of Pepper Plants

When plants suffer from oxidative stress caused by a wide range of environmental factors, the increased activities of SOD, CAT, and POD in the antioxidant enzyme system of plants can effectively prevent active oxygen accumulation, reducing the MDA content and enabling the plants to grow normally [41]. Regarding light quality as an environmental factor, researchers have explored its effects on the plants' antioxidant enzyme system in recent years. For instance, with the supplementation of blue light, the antioxidant enzyme activities increased in onion plants [42]. Cotton plants exposed to the light exclusion of UV-B/A had lower levels of antioxidant enzyme activities [43]. In the present study, examination analysis in pepper seedlings revealed that in comparison with control, blue light had no significant effect on the examined antioxidant enzyme activities and MDA content, UV-A light merely induced a decrease in the SOD activity, indicating that enhanced blue and UV-A light proportion did not cause oxidative stress on pepper seedlings (Figure 3). Exposure of plants to supplementary FR light led to significantly higher antioxidant enzyme activities and greater stress resistance [3,44]. In this present study, when adding far-red proportion to pepper seedlings, compared with the control, the CAT activity was significantly increased in the FR30 seedlings, and significantly lower MDA content occurred in the FR20 and FR30 seedlings, meaning that supplementary with far-red light might improve the antioxidant capacity of pepper seedlings (Figure 3).

The flowering and fruiting performance after transplanting are largely influenced by the growth and development of seedlings [45,46]. In this study, the flowering time of pepper seedlings supplemented with blue, UV-A, and far-red light was not significantly different from that of the control after transplanting (Table 4). Supplemental blue light and far-red light in the whole developmental period delayed and advanced the flowering time, respectively [47,48]. Different light quality treatments carried out only in the seedling stage of pepper plants might result in insufficient light dose to make a difference in the flowering time in this experiment.

Concerning fruit development, although the average fruit weight per plant was not significantly altered in B30 and B60 treated pepper plants, the average fruit amount per

plant produced by pepper seedlings in B60 was the highest (Table 4). The average number of fruits was not significantly affected by the elevated proportion of UV-A light at seedling stage, but UV-A treated pepper plants showed increases in the average weight of fruits, which was significantly elevated in the UVA6 treatment (Table 4). These results suggest that when cultivating pepper seedlings in a plant factory, supplemental UV-A and blue light helped to improve the early yield of pepper plants after transplanting, consistent with the results of Son et al., who observed significantly elevated fruit yield per plant after transplanting of tomato seedlings grown under an appropriately increased blue light ratio [49].

## 5. Conclusions

In this study, we investigated the effects of LED light combined with different light qualities (CK, B30, B60, UVA2, UVA6, FR20, and FR30) on the growth and development of pepper seedlings, and the subsequent performance of these different pepper seedlings after transplanting. Briefly, supplemental blue or UV-A light (B30, B60, UVA2, and UVA6 treatments) could increase the seedling index of pepper seedlings and pepper fruit yield after transplantation, while supplemental FR light (FR20 and FR30 treatments) did not. Overall, our results demonstrated that a supplemental proportion of blue and UV-A light could be used as a strategy to enhance the quality of pepper seedlings in a PFAL and improve their performance after transplanting.

**Supplementary Materials:** The following supporting information can be downloaded at: https://www.mdpi.com/article/10.3390/agronomy12102269/s1.

**Author Contributions:** X.L., R.S., and M.G., performed the experiments and wrote the manuscript. R.H. and Y.L. performed the experiments and statistical analyses. H.L. conceived and designed the experiments. H.L. acquired of funding and contributed to revised the manuscript. All authors have read and agreed to the published version of the manuscript.

**Funding:** This study was supported by the National Key Research and Development Program of China (2017YFE0131000), and the Key Research and Development Program of Ningxia (2021BBF02024).

**Data Availability Statement:** The data that support the findings of this study are available from the online at https://www.mdpi.com/ethics (accessed on 17 September 2022).

**Conflicts of Interest:** The authors declare no conflict of interest.

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
