# Peer review of "Effects of LED Light Quality on the Growth of Pepper (Capsicum spp.) Seedlings and the Development after Transplanting"

_agronomy, doi:10.3390/agronomy12102269_

Round 1
Reviewer 1 Report
The subject of the MS "Effects of LED light quality on growth and development of pepper seedlings in a plant factory" is of current interest, however there are a lot of research results on the effects of light quality on plant growth and development have been reported already. Therefore, in this MS the novelty of the reseach should be described more clear with the emphasis on what was not known before. If the novelty is in the effects of different light qualty trratments on the pepper yield (in greenhouses), then it is not clear why the title is about "seedlings" and "plant factory" only. The MS needs substantial improvement.
Major concerns
Title. There ia a discrepancy between "seedlings - plant factoty" in the title and "flowering, yield - greenhouse" in the text.
Introduction. There is no a word about the effects of FR-enriched lighting on plant frowth. However, this information is available in the literature.
English should be carefully revised throughout the manuscript to improve the comprehension. For examle: lines 14-15 - height cannot be promoted ot inhibited. The growth can be; Line 17 - chl content of the treatment - the treatment has no chl? plants have; Line 21, 135 - seedlings were (not was); Fig 2E - true (not ture); Line 221 - there (not the) et al. Throughout the MS, please check the use of single and plural 'index' and "indices". At present their use is very confusing.
In the Methods and Materials there is no description of the method used for measuring chlorophyll content (what spectophotometer was used?).
Line 101: MINI-PAM II is a Photosynthesis Yield Analyzer, or fluorimeter (not apparatus machine)
Line 102: Y must be measured on light-adapted leaves, Fv/Fm - on dark adapted. Please, explain how did you measure Y on dark-adapted leaved.
Fig. 2 - the quality of text in the axis titles has to be improved
Discussion should be better structured. Please, try to explain some discrepancies with results obtained in other research. For example, it was shown that FR-enriched lighting affects plant growth and morphology, but responses can greatly vary depending on light intensity and plant species. In particular, the growth of red pepper was positively correlated to far-red light at certain intensities (Hwang et al., 2020). Therefore, if you make a conclusion that supplemental FR light could decrease pepper seedling indexes, try to explain why the results differ. Anyway, the conclusion is too broad. Plase, emphasize what new knowledge your research added to already known facts that blue light is beneficial for plant growth (biomass accumulation) et al.
Minor concerns.
Line 58: What is total light intensity? Whay total?
Line 211: electron transport rate (not efficiency)
Table 3 - in the titlae - values (not measurements).
Tables 3, 4. Units are lacking.
There are data on chl a and chl b content in the table, but they are not discussed anyhow. Please, consider to present the ratio chl a/b intead.
Line 239: 'growth of the flowers" - it was not studied.
Line 297: What is plant type?
Throughout the manuscript (references) latin nameas should be italicized (lines 372, 381, 392, 396, 403,416,422, 440,444, 446), change capital letters to small ones on lines 403, 415, 437, 446.
Author Response
Dear Editor,
Thank you for your kind comments and valuable suggestions, as well as those from the reviewers. We have carefully revised the manuscript (agronomy-1919671) according to each of these suggestions. Enclosed, please find the latest manuscript and our replies to the comments. We trust our combined effort has improved our manuscript significantly. To show clearly the difference, we have used the "Track Changes" function in Microsoft Word. Thank you for providing us with the opportunity to revise our manuscript. We sincerely hope this manuscript will be finally acceptable to be published on Agronomy. Thank you very much for all your help and looking forward to hearing from you soon.
Sincerely,
Xiaojuan Liu
College of Horticulture
South China Agricultural University
Guangzhou, Guangdong 510642

Reviewer 2 Report
The manuscript by Liu et al., "Effects of LED light quality on growth and development of pepper seedlings in a plant factory" (agronomy-1919671), demonstrated the pepper (Capsicum) a commercial plants cultivate by different light qualities in a farming system. The author, demonsstre many analysis growth and development, biochemical analyses and a relative good presentation in this manuscript.
The authors have done a large amount of work, employing various references and critical analysis based on a scientific method and structure. The introduction its contextuaized and Ok. However, many section, is necessary adjusts. Material and methods its necessary improve some rephrase and reference methods. For example, the enzymatic method (CAT, POD, SOD) is poor describe in manuscript. In addition, statistica description is also necessary. Please, carefully all check that section.
Results section needs major revision, as wee as discussion. The comparisons described are often confusing. Some discussion phrases are present in the results section. The discussion is small and needs to be improved. However, it is an interesting manuscript, with robust data and that fits the Agronomy journal's scopus, requiring modifications only in the manuscript. I think that after the major modifications, it can be accepted for publication.
Figures and tables need major adjustments or reformulation. Please, check my all comments in pdf attached file.
In addition, a attached a pdf file, with major corrections to English Grammer, suggesting and questions which its not clearance for me. Please, check in manuscript American or British English by Native Expert ou your Collegue Native. Please, check my all consideration in your manuscript.
Best regards

Author Response

(The authors gave the same response as above.)

Reviewer 3 Report
The manuscript discusses the effects of various LED light on growth and development of pepper. Technical content, flow, and comparison of the work with other relevant works are all appropriately provided in the manuscript. Hence, I see the manuscript qualified for being accepted in Agronomy MDPI . However, before acceptance, the manuscript needs some improvements:
Below are some examples where improvements are needed
· I suggest in the introduction, the authors cite some of the recent relevant research where the effect of light on enhancing the growth and development of agriproducts have thoroughly been addressed such as
a) Recent applications of novel laser techniques for enhancing agricultural production. Laser Physics, 31(5), p.053001.
b) Effect of laser biostimulation on germination of wheat. Applied Engineering in Agriculture, 38(1), pp.77-84.
· Line 70-72, confusing. It needs to be re-written
· What are the model number and manufacture of the light sources? More info is needed.
· Line 84: hypocotyl length was measured by using a tape. How? Do you have a reference?
· Line 90: thousandth electronic balance. What is the model number and manufacturer?
· Line 94-97: the equations should be better presented. Maybe 1 equation per line??
· Line 281-283 need to be rewritten.
· The image qualities are very low. Can you improve them?
Author Response

(The authors gave the same response as above.)

Round 2
Reviewer 1 Report
The MS has been improved significantly, but there are still some shortcomings.
1. The new title is not good as the word “their” is not clear – effects of light quality (then why plural “their”)?
2. Why do you use “total light intensity”? It is just “light intensity of 200 μmol m−2s−1.”.
3. Table 3: Chl a and b – a and b should be italicized.
4. Line 323: the year for the ref. Hwang et al. is missing.
5. Lines 365, 368: not “pigment is”, but “pigments are”.
6. Lines 381-384: Please, read the theory. Parameters of chl a fluorescence are not connected with carotenoid content (you write photosynthetic pigment content) and
7. Conclusion is very broad: supplemental blue or UV-A light (please, specify treatment) could increase the seedling index of pepper seedlings and pepper fruits yield after transplantation, while supplemental FR light (please, specify treatment) did not.
Author Response
Dear Editor,
Thank you for your kind comments and valuable suggestions, as well as those from the reviewers. We have carefully revised the manuscript (agronomy-1919671) according to each of these suggestions once again. Enclosed, please find the latest manuscript and our replies to the comments. We trust our combined effort has improved our manuscript significantly. To show clearly the difference, we have used the "Track Changes" function in Microsoft Word. Thank you for providing us with the opportunity to revise our manuscript. We sincerely hope this manuscript will be finally acceptable to be published on Agronomy. Thank you very much for all your help and looking forward to hearing from you soon.
Sincerely,
Xiaojuan Liu
College of Horticulture
South China Agricultural University
Guangzhou, Guangdong 510642
E-mail: 276442487@qq.com
Corresponding author:
Name: Houcheng Liu
E-mail: liuhch@scau.edu.cn

Reviewer 2 Report
I would like to thank the authors for addressing my comments. I consider the authors made important changes in the manuscript and it was highly improved. I recommend the publication of the manuscript in its current form. Best regards.
Author Response
Dear Editor,
Thank you for your kind comments and valuable suggestions, as well as those from the reviewers. We have carefully revised the manuscript (agronomy-1919671) according to each of these suggestions. Thank you for providing us with the opportunity to revise our manuscript. We sincerely hope this manuscript will be finally acceptable to be published on Agronomy. Thank you very much for all your help and looking forward to hearing from you soon.
Sincerely,
Xiaojuan Liu
College of Horticulture
South China Agricultural University
Guangzhou, Guangdong 510642
E-mail: 276442487@qq.com
Corresponding author:
Name: Houcheng Liu
E-mail: liuhch@scau.edu.cn
